# Nutrients and Dietary Patterns Related to Osteoporosis

**DOI:** 10.3390/nu12071986

**Published:** 2020-07-03

**Authors:** Araceli Muñoz-Garach, Beatriz García-Fontana, Manuel Muñoz-Torres

**Affiliations:** 1Department of Endocrinology and Nutrition, Virgen de las Nieves Hospital, 18014 Granada, Spain; 2Instituto de Investigación Biosanitaria (Ibs.GRANADA), 18014 Granada, Spain; bgfontana@fibao.es; 3CIBERFES, Instituto de Salud Carlos III, 28029 Madrid, Spain; 4Unidad de Gestión Clínica Endocrinología y Nutrición, Hospital Universitario San Cecilio de Granada, 18016 Granada, Spain; 5Department of Medicine, University of Granada, 18016 Granada, Spain

**Keywords:** osteoporosis, calcium intake, vitamin D, dairy products, protein, dietary patterns

## Abstract

Osteoporosis is a common chronic disease characterized by a decrease in bone mineral density, impaired bone strength, and an increased risk of fragility fractures. Fragility fractures are associated with significant morbidity, mortality and disability and are a major public health problem worldwide. The influence of nutritional factors on the development and progression of this disease can be significant and is not yet well established. Calcium intake and vitamin D status are considered to be essential for bone metabolism homeostasis. However, some recent studies have questioned the usefulness of calcium and vitamin D supplements in decreasing the risk of fractures. The adequate intake of protein, vegetables and other nutrients is also of interest, and recommendations have been established by expert consensus and clinical practice guidelines. It is important to understand the influence of nutrients not only in isolation but also in the context of a dietary pattern, which is a complex mixture of nutrients. In this review, we evaluate the available scientific evidence for the effects of the main dietary patterns on bone health. Although some dietary patterns seem to have beneficial effects, more studies are needed to fully elucidate the true influence of diet on bone fragility.

## 1. Introduction

Osteoporosis is a disabling disease that results in fragility fractures, causes high morbidity and mortality, and increases healthcare costs [1]. The promotion of healthy habits is very important for reducing the risk of osteoporosis. Ensuring an adequate dietary intake of calcium, vitamin D and protein—as well as performing regular weight-bearing exercise and abandoning harmful habits, such as alcohol intake and smoking—helps to improve bone quality. Other nutrients seem to have additional influence. European guidance for the diagnosis and management of osteoporosis in postmenopausal women recommends a daily intake of at least 1000 mg/day of calcium, 800 IU/day of vitamin D to maintain serum 25-hydroxyvitamin D levels >50 nmol/L and 1 g/kg body weight of protein for all women aged over 50 years for the prevention of the age-related deterioration of musculoskeletal health [2]. Although drug therapy appears to be the first-line option for reducing the risk of fractures in the elderly, it is not always feasible, and dietary modifications—specifically, increasing calcium, vitamin D and protein intake—may be a more pragmatic option [3]. For the present article, we have performed an evidence-based review of the most relevant publications available to date associating bone health with nutrition.

## 2. Osteoporosis and Related Nutrients

### 2.1. Calcium, Vitamin D and Dairy Products

Calcium and vitamin D form part of the bone mineral matrix as calcium phosphate (hydroxyapatite crystals) and are required for bone strength. The best way to achieve adequate calcium intake is through adhering to a healthy diet. However, sometimes dietary sources are insufficient or poorly tolerated, and, in those situations, pharmacological calcium supplementation could be useful. This is the recommendation of most clinical practice guidelines [4,5,6,7]. However, some authors are critical of this recommendation due to its low efficacy and the possibility of adverse effects [7]. The most important sources of calcium in the diet are dairy products (milk, yogurt and cheese), fish (especially sardines with bones), pulses and a few vegetables and fruits (particularly nuts and seeds).

Calcium homeostasis is, predominantly, regulated by vitamin D. A total of 80–90% of vitamin D is obtained from cutaneous synthesis after sunlight exposure, and 10–20%, from a limited number of foods, such as oily fish, mushrooms and some fortified dairy products. Nevertheless, no foods can provide enough vitamin D to meet requirements. There is a potential role for the fortification of different foods to increase vitamin D intakes.

Adequate sunlight exposure is important for preventing and correcting vitamin D insufficiency. Vitamin D deficiency has deleterious consequences for health outcomes, and this vitamin is extremely important for bone health maintenance. In elderly or postmenopausal women, vitamin D deficiency may aggravate osteoporosis. Additionally, adequate concentrations of 25-hydroxyvitamin D (25(OH)D) are necessary for maximizing the efficacy of anti-osteoporotic drugs [8]. Furthermore, vitamin D is involved in maintaining muscle mass and strength as well as bone structure. High efficacy has been demonstrated for certain vitamin D-fortified foods, such as reduced-fat cheese and vitamin D-biofortified eggs [9,10].

A recent meta-analysis of the impact of vitamin D-fortified foods on serum 25(OH)D levels, bone mineral density (BMD) and bone turnover markers (BTM) showed significant effects, with an increase in serum 25(OH)D and BMD and decrease in parathormone (PTH) levels. No significant beneficial effect on BTM was found [11].

Dairy products are the main food sources providing bone-beneficial nutrients, such as calcium, phosphorus and magnesium and these elements have a morphological role in bone healthy structure. Dairy products are also a source of protein, vitamin B-12, zinc, potassium and riboflavin. Dairy products could also be considered as a remarkable source of vitamin D if fortified with such. Vitamin D concentrations based on weight are higher in yogurt and cheese than in milk, but the serving sizes are typically smaller for these products than for milk. Table 1 shows the main nutrient content per 100 g of commonly consumed foods.

Therefore, the regular consumption of plain dairy products or those fortified with calcium and/or vitamin D may increase the total body bone mineral content (BMC). However, this association varies depending on ethnicity. In Caucasian and Chinese women, calcium from (fortified) dairy products increased BMD by 0.7–1.8% over 2 years, depending on the site of measurement (lumbar spine, total hip, femoral neck or total body). In adult Caucasian women, a daily intake of 200–250 mL of milk has been associated with a 5% or higher reduction in fracture risk [12]. Another large prospective study recently showed a significant 8% reduction in the risk of hip fracture for each daily serving of milk, in men and women combined, which was evaluated every 4 years. Moreover, milk intake was shown to reduce the risk of osteoporotic fractures by 5–15% in an individual patient data cohort, especially in older patients (80 years or more) [13]. In a cohort of postmenopausal healthy women, the daily consumption of milk fortified with calcium and vitamin D significantly improved vitamin D status and BMD at the femoral neck, as well as having favorable effects on glucose and lipid profiles [14]. The data on this effect in males are more limited.

Fermented milk products, such as yogurt or soft cheese, are also an important source of calcium, phosphorus, and protein. Fermented milk is interesting because it may contain prebiotics and provide probiotics. For example, inulin (a prebiotic) may be added to yogurt. Furthermore, probiotics are able to improve intestinal calcium absorption and bone metabolism. Bone mass balance and the attenuation of sex hormone deficiency-induced bone loss (especially important after menopause) seemed to improve after the intake of calcium, protein, prebiotics and probiotics. These compounds may change the gut microbiota composition and metabolism. Fermented milk product consumption might also be considered as a marker of healthy lifestyle promoting bone maintenance [15].

In animal models, probiotics have been shown to inhibit the bone loss associated with estrogen deficiency, diabetes, or glucocorticoid treatments, by regulating both bone resorption and formation [16]. In humans, probiotics alter 25(OH)D levels and calcium absorption, and slightly decrease bone loss in elderly postmenopausal women to a similar extent as vitamin D supplements with or without calcium combination [17]. Fermented dairy products are an important dietary source of probiotics, which can improve calcium homeostasis, prevent secondary hyperparathyroidism, and inhibit age-related increases in bone resorption and bone loss. Further studies are needed to establish whether probiotics or any other intervention targeting the gut microbiota and its metabolites may be considered as adjuvant treatments for use alongside calcium and vitamin D supplementation or anti-osteoporotic drugs for the management of patients with fragility and low bone mass [18].

Unfortunately, a low calcium intake and a suboptimal vitamin D status are very common in the elderly in Europe. The daily consumption of calcium and vitamin D-fortified food products (most commonly yogurt or milk) can help to improve vitamin D intake [19]. Yogurt consumption may guarantee a more regular ingestion of dairy products and higher adherence to their consumption, because of various flavors and sweetness levels. However, not enough studies addressing bone remodeling markers, other than BMD or BMC, allowing a good comparison between the effect of dairy products and other food sources on bone quality are available.

Not all of the available information supports the idea of a high intake of dairy products being beneficial for bone [20]. In the US, data from the NHANES III population-based survey, including a cohort of nearly 10,000 women and men, showed no correlation between calcium intake and BMD at the hip site, and the correlation was more applicable for subjects with higher levels of 25(OH)D [21]. Only women with higher 25(OH)D levels seemed to benefit from a higher calcium intake. Other meta-analyses based on prospective studies were not able to associate a higher milk intake or total dairy product consumption with the risk of hip fracture in women or men [22,23]. The Study of Women’s Health Across the Nation (SWAN) recently tried to demonstrate the benefits of dairy intake for bones, but its results were not positive for long-term bone health. It is important to mention that this study was a multicenter, multi-ethnic, community-based longitudinal cohort project that included women during their middle years around the menopause transition period [24]. The benefit of dairy consumption for preserving BMD or preventing fractures could not be established. However, when interpreting these results, the authors reported that dairy intake was low among SWAN participants, and this fact could have influenced their results.

The potential beneficial role of dairy products in the prevention of fractures is controversial [25]. Moreover, a higher consumption of milk or yogurt could be related with a lower risk of hip fracture although with some differences between populations with different characteristics [26]. The inconsistent results found regarding the possible association between milk consumption and the risk of hip fracture could be partially explained by the differences in policies about vitamin D fortification across different countries.

In conclusion, the beneficial role of dairy products in bone health has been more extensively established in Chinese and Caucasian girls and women [27], but these results could not be replicated for other populations. Nevertheless, the available evidence is insufficient because studies on calcium alone and other dairy products are limited in size, number, and duration and advanced research on the influence of dairy products within the context of bone health-promoting diets in specific ethnic groups, other than the above-mentioned studies, is needed [27].

In summary, it is very difficult to differentiate between the effects of dairy intake and those of other components of diet and lifestyle. No large randomized clinical trials evaluating dairy intake over long periods of time are available. Therefore, we must assume this limitation and establish our recommendations based on the best observational studies and on the previously known biological premises.

### 2.2. Other Minerals

Other minerals, such as potassium or magnesium, also have an important role in bone health.

Dietary potassium may reduce the acid load and, thus, calcium depletion from bones. Apart from its role in the maintenance of an alkaline state in the body, potassium can also increase the accumulation of calcium in the kidneys. In a nationwide Korean population study, the highest intake of potassium was associated with greater lumbar, total hip, and femur neck BMD in men older than 50 years and postmenopausal women [28]. This association was demonstrated in a population with a low calcium intake. The meta-analysis performed by Lambert et al. confirmed that supplementation with alkaline potassium salts was associated with a significant reduction in renal calcium excretion and acid excretion. These salts significantly lowered the bone resorption marker cross-linked N-telopeptides of type I collagen (NTX) but with no effect on bone formation markers or BMD. The reduction in bone resorption indicates a potential advantage for bone health [29].

Magnesium is also necessary for calcium metabolism [30]. After potassium, magnesium is the second most abundant intracellular cation, with a concentration of 10–30 mM in the human body. Magnesium is found in most whole foods, such as green leafy vegetables, legumes and nuts. The recommended daily allowances of magnesium are 310–360 mg and 400–420 mg for women and men, respectively. The requirements vary between individuals according to age, sex and previous nutritional status [31].

Magnesium is also involved in the exchange of calcium and potassium ions across cell membranes, and it is essential for neuronal activity and muscle contractions. Approximately 50–60% of the total body magnesium content is accumulated in the bone. In the bone’s structure, magnesium ions bind at the surface of the hydroxyapatite crystals, improve the solubility of phosphorous and calcium hydroxyapatite and, thereby, influence crystal size and formation. Furthermore, magnesium induces osteoblast proliferation; therefore, its deficiency is associated with reduced bone formation [32]. Magnesium is also necessary for the activation of vitamin D because most of the enzymes involved in vitamin D metabolism require magnesium [33].

Populations that consume more processed food (more refined grains, sugars and fats) have lower levels of magnesium, as is typical in the United States [34]. A significant problem arises from the soil used for agriculture, which is becoming increasingly deficient in essential minerals such as magnesium.

No randomized studies evaluating the effect of magnesium on bone disease are available, but small-size studies have associated low serum magnesium levels with osteoporosis [35,36]. In older subjects, a low intake of magnesium causes excessive calcium release from the bone, which further worsens bone fragility and increases the risks of fractures and falls [37]. A cross-sectional analysis of a cohort of subjects in the UK evaluating the influence of dietary magnesium and potassium intakes on bone density and fracture risk found a lower hip fracture risk in both men and women with a higher intake of these minerals, but these results did not hold after adjustment for multiple testing [38]. Figure 1 summarizes the main influence of some important nutrients on bone health.

### 2.3. Protein Intake

Protein intake is also essential for bone health. Approximately 50% of bone volume and about a third of bone mass is composed of proteins. They are integrated into the organic matrix of bone as part of the collagen structure when mineralization occurs. Dietary proteins also affect the secretion and action of insulin-like growth factor I (IGF-I), an orthotropic hormone important for bone formation. The IGF-I hormone improves calcium and phosphorus absorption in the gut; it is involved in the synthesis of calcitriol, and it increases the rate of phosphate reabsorption from the kidney. An adequate amount of dietary protein is therefore required for the maintenance of bone health. The European Society for Clinical and Economic Aspects of Osteoporosis and Osteoarthritis (ESCEO) recommends a dietary protein intake of 1.0–1.2 g/kg body weight/day, with at least 20–25 g of high-quality protein at each main meal [39]. The main sources of protein in healthy diets come from meat, fish, poultry, eggs and dairy products. Adequate protein intake is imperative for bone matrix formation and maintenance. However, it was previously believed that excessively high protein intake might induce a negative calcium balance, based on the “nutritional acid load hypothesis”. This hypothesis relates a higher protein intake (especially that of animal origin, with more sulfur amino acids) to increased acid production and bone resorption, and thus, the development of deleterious hypercalciuria and bone loss, leading to osteoporosis and an increased risk of fragility fractures [40]. More recent meta-analyses have reported that a higher protein intake—more than 0.8 g/kg body weight/day, higher than the general dietary recommendations—is associated with a higher BMD, reduced risk of hip fracture and slower rate of bone loss. It is especially important in older people with osteoporosis and must always be accompanied by adequate dietary calcium intakes [41]. It is also important to take into account the anabolic effect of protein intake that, together with physical activity, is a main driver of muscle protein synthesis. When practicing exercise, muscle mass and strength improve, and the combination of adequate protein intake and exercise induces a greater degree of muscle protein accretion than either intervention alone. In the same way, a balanced dietary protein intake in combination with resistance exercise is an important contributor to the maintenance of bone strength.

It has been demonstrated that there is no adverse effect of higher protein intakes on bone. Furthermore, they are beneficial in attenuating age-related bone loss and reducing hip fracture risk in older subjects [42]. No significant differences in hip fracture outcomes between animal and vegetable protein consumption within the limits of a balanced diet have been found. However, as previously mentioned, all of these positive outcomes depend on adequate calcium intakes. Dairy product intake is probably very helpful because these products are a source of both proteins and calcium, since 1 L of milk provides 32 g of protein and 1200 mg of calcium. Furthermore, in some countries, yogurts are supplemented with milk powder, resulting in a 50% increased content of these nutrients compared with that in yogurt prepared from plain milk. The combination of protein and calcium in dairy products has positive effects on calciotropic hormones. They produced a reduction in circulating PTH, an increase in IGF-I and, consequently, a decrease in bone resorption markers, as well as improving BMD [15]. Therefore, much more severe problems are found for diets with inadequate protein than for those with an excess.

In terms of the nutrients required for good bone health, fruit and vegetables are rich in the previously mentioned nutrients, such as potassium and magnesium, and in vitamin C, vitamin K, folate and carotenoids [44,45].

### 2.4. Vitamins K and C

Vitamin K is involved in bone matrix formation during mineralization. It acts as a cofactor for the microsomal γ-carboxylase, which facilitates the post-translational conversion of glutamyl to γ-carboxyglutamyl residues in osteocalcin, and it also influences other vitamin K-dependent proteins. In its γ-carboxylated state, osteocalcin is a calcium-binding protein in bone, facilitating the mineralization process. Vitamin K comprises a family of different molecular forms: vitamin K1 is a single form synthesized by plants, whereas the group of vitamins K2 comprises multiple forms, mainly synthesized by bacteria. Vitamin K1 is the major type found in human diets. A special feature of cheese is the presence of vitamin K2. A recent meta-analysis performed in 2019, including more than 11,000 patients, was conducted mainly in postmenopausal or osteoporotic patients [46]. It concluded that vitamin K supplementation appeared to have little clinically significant effect on BMD and vertebral fracture outcomes for these patients—in part, related to the heterogenicity in studies included, especially concerning the treatment regimes. Regarding postmenopausal women and osteoporotic patients, clinical fractures were less frequent in the vitamin K-supplemented group, but the effect appeared to be smaller when the analysis was restricted to low-risk-of-bias studies. Unfortunately, very few trials are available; therefore, it is difficult to draw conclusions for other populations based on this meta-analysis. Another systematic review found that the clinical use of oral vitamin K antagonists as part of anticoagulant therapy was neither related to nor reduced BMD, nor was it associated with an increased risk of fracture [47]; this supported previously reported results suggesting the absence of a clinically meaningful effect of vitamin K on BMD.

Vitamin C can improve bone health because of its antioxidant properties. It is able to suppress osteoclast activity [48]. It also acts as a cofactor for osteoblast differentiation and participates in collagen formation. Vitamin C is a marker of a healthy dietary pattern rich in fruit and vegetables. A systematic review and meta-analysis that compiled observational studies concluded that a greater dietary vitamin C intake was positively associated with BMD at the femoral neck and lumbar spine, although this review reported remarkable between-study heterogeneity at the femoral neck. This heterogeneity was caused by differences regarding study design, sex and age. A higher dietary vitamin C intake was correlated with a lower risk of hip fracture and osteoporosis, as well as higher BMD, at both the femoral neck and lumbar spine sites [49]; a more recent meta-analysis strongly reinforces the idea that increasing dietary vitamin C intake could decrease the risk of hip fractures in both males and females [50]. Although these bone benefits are remarkable, there is little information in clinical practice guidelines in terms of vitamin C intake recommendations, and this should be taken into account.

### 2.5. Omega-3 Polyunsaturated Fatty Acids and Other Nutrients

There is conflicting evidence regarding the effects of omega-3 polyunsaturated fatty acids (PUFA) on bone metabolism. The consumption of eicosapentaenoic acid (EPA) and docosahexaenoic acid (DHA) may influence bone growth and remodeling in humans through the inhibition of bone resorption and also by stimulating bone formation [51]. The mechanism by which PUFAs may affect bone turnover is not well known, but it is hypothesized that both EPA and DHA may exert their benefits by regulating osteoprotegerin (OPG)/receptor activator of nuclear factor kB (RANK), with a balance towards bone formation [52]. Fish and seafood are rich in PUFAs, especially n–3 FAs, which are known to have an anti-inflammatory effect that improves bone quality [53]. A meta-analysis conducted by Shen et al. showed that omega-3 fatty acids reduced osteocalcin serum levels in postmenopausal women, but no significant decrease in bone-specific alkaline phosphatase was found [54]. A study performed with a dairy product enriched in PUFAs, calcium, oleic acid and vitamins to evaluate their effects on bone metabolism in postmenopausal women showed favorable changes in some bone metabolism markers, such as an increase in vitamin D levels and a decrease in both PTH and RANKL, but did not show changes in other bone turnover markers or serum OPG [55]. Another Spanish study showed that the dietary intake of PUFAs was positively associated with BMD in both healthy and osteopenic Spanish women, at both the hips and lumbar spine [56]. A systematic review published in 2019 revealed a significant inverse association between the dietary intake of n-3 PUFAs through fish consumption and the risk of hip fracture, and that they might have protective effects on bone health [57].

Folate and vitamin B-12 might also influence bone by reducing homocysteine concentrations; homocysteine is linked to lower BMD and a higher risk of fracture [58]. A recent meta-analysis including 7475 individuals from four prospective studies showed a 4% lower fracture risk for each 50 pmol/L increase in vitamin B-12 concentration [59].

Zinc can be found in beans, nuts and whole grains, but the phytate in these foods makes it less bioavailable than zinc from animal-based sources. Lower serum and bone zinc concentrations have been described in patients with osteoporosis [60].

## 3. Recent Studies on Reference Dietary Patterns

Dietary pattern approaches unify contributions from various aspects of the diet. They are more useful than single-nutrient and food studies for elucidating the effects of different nutrients alone on bone health and can paint a more comprehensive picture.

In general, a dietary pattern with a high intake of fruit, vegetables, low-fat dairy products, whole grains, poultry, fish, nuts and legumes has been demonstrated to have a positive effect on bone health and directly associated with a better BMD and lower risk of fracture. Moreover, it is inversely associated with levels of bone resorption markers [61,62]. In an Asian population, a dietary pattern including higher intake of fruits, vegetables and soy was associated with a lower risk of fractures and also a lower risk of osteoporosis [63].

### 3.1. Mediterranean Diet

The latest studies have shown that adherence to the Mediterranean diet (Med-Diet) is protective against osteoporosis [64]. Animal studies have suggested that antioxidant-rich fruits have a marked effect, increasing trabecular bone volume, number, and thickness, and decreasing trabecular separation through the stimulation of bone formation and suppression of bone resorption [65].

Those subjects with a higher adherence to Med-Diet showed an independent association, with an increase in 25(OH)D, suggesting that higher vitamin D levels could mediate the protective effect of the Med-Diet against osteoporosis. In postmenopausal women, it has been reported that higher Med-Diet scores were associated with a higher BMD and lower risk of hip fracture [66,67]. In a cohort of Spanish premenopausal women, the evaluation of BMD showed that all, total, trabecular, and cortical bone density were positively associated with higher adherence to the Med-Diet [68]. These positive bone results were also demonstrated in a cohort of Italian subjects, where higher T-scores were positively associated with higher adherence to the Med-Diet [69]. In another study including postmenopausal women, those with higher adherence to Med-Diet had better BMD at the lumbar spine and improved muscle mass, also important for the prevention of osteoporosis and fractures [70]. Greater adherence to this diet was associated with a lower risk of fracture as well as a higher mean BMD, according to the meta-analysis of four effect sizes, obtained from three studies by Malmir et al. [71].

Considered to be at the core of the Mediterranean diet, olive oil has been reported as beneficial for bone status. Published evidence suggests that olive oil, with a high proportion of phenols, can be beneficial by preventing the loss of bone mass [72]. It has been demonstrated that the phenolic compounds can modulate the growing capacity and cell maturation of osteoblasts by increasing alkaline phosphatase activity and contributing to the creation of the extracellular matrix. The dietary intake of olive oil was significantly associated with higher BMD in a cohort of Spanish women across a wide range of ages [73]. Furthermore, it was associated with a higher total osteocalcin concentration and an increase in procollagen I N-terminal propeptide in a group of men, suggesting the protective effects of olive oil on bone [74]. A cohort of subjects was analyzed as part of the PREDIMED study, and those participants with the highest intake of extra-virgin olive oil had the lowest risk of osteoporosis-related fractures [75].

### 3.2. Western Diet

Regarding the situation in Europe, as suggested by the “Framingham study”, individuals with diets high in processed protein foods (with a high percentage of protein intake from cheese, processed meat, pastries, pizza, French fries, snacks and refined grains) have shown a lower BMD compared to other groups of subject [76]. Furthermore, all studies alert about the risks of the unhealthy Western dietary pattern and the importance of promoting a decrease in the consumption of processed food products, sweets and desserts, soft drinks, fried foods, meat and refined grains. The risks of Western diets should be reinforced, as they are associated with lower BMD and a higher risk of fractures [77]. As an example, an English traditional dietary pattern—including a high intake of fried fish and potatoes, legumes, red and processed meats, and savory pies—was inversely associated with BMD in the femoral neck [78]. Moreover, a high fat intake, derived mostly from refined carbohydrates and fat products, can directly interfere with intestinal calcium absorption and also increase fat accumulation and obesity, which lead to a decrease in osteoblast differentiation and bone formation [79]. Sodium intake induces higher calciuria, which is presumed to increase bone loss and bone remodeling [80]. An excessive intake of inorganic phosphorus, present in processed food additives, induces a disruption of the calcium–phosphorus ratio, affecting the endocrine regulation of calcium homeostasis [81]. This is deleterious for bone health. However, it might be taken into consideration that this detrimental relationship between the Western dietary pattern and bone health is, in part, confounded by high net endogenous acid production. In an acidic environment, bone acts as a provider of alkali to maintain the acid–alkali balance, which leads to progressive bone loss [82].

### 3.3. Asian Diet

Asian populations have dietary patterns in which soy and fish intake are high compared to that in Western populations. They have a significantly lower incidence of osteoporotic fractures. Indeed, several meta-analyses have demonstrated that the supplementation of soy isoflavones with omega-3 fatty acids improved bone health status in women [83]. Evidence from epidemiological studies supports the idea that dietary soy isoflavone intake attenuates the bone loss induced by menopause, improving bone formation and decreasing bone resorption [84]. Another study in a Korean population showed that an increased consumption of fruits and dairy products as part of the traditional Korean diet, consisting mainly of white rice and vegetables, might decrease the risk of osteoporosis in Korean postmenopausal women [85] compared to the consumption of a diet rich in meat, alcohol and sugar. Furthermore, those subjects with higher intakes of white rice, kimchee and seaweed had a higher risk of osteoporosis in the lumbar spine. Another study in a cohort of Japanese postmenopausal women found that natto (fermented soybeans) consumption was positively associated with lumbar spine BMD, and its intake could be recommended for preventing postmenopausal bone loss [86].

A systematic review of dietary patterns involving a higher consumption of meat or processed meat (Meat Diet) or a higher consumption of fish and seafood (Fish Diet) suggested that these diets did not alter BMD or the risk of fractures. This study indicated that protein intake from fish or meat is not harmful to bone. The Meat Diet showed negative effects on bone metabolism in the setting of a Western Diet, but these could not be demonstrated in the context a Mediterranean or Asian Diet [87].

### 3.4. Vegetarian Diets

Vegetarian diets have been proven to contain lower amounts of calcium, vitamin D, vitamin B-12, protein and n–3 fatty acids, all of which have important roles in maintaining bone health. However, healthy vegetarian diets usually contain greater quantities of several protective bone-related nutrients such as magnesium, potassium, vitamin K, and antioxidant and anti-inflammatory phytonutrients. Unfortunately, the limited available evidence suggests that, on balance, vegetarians—especially vegans—may be at higher risk of low BMD and fractures [88]. A meta-analysis in 2009 by Ho-Pham comparing vegetarians and omnivores, including more than 2500 subjects, showed a 4% lower BMD at both the femoral neck and lumbar spine in vegetarians than in omnivores [89]. From this study, a subgroup analysis found that the difference was greater for vegans, who had a 6% lower BMD than omnivores.

There are few studies available examining vegetarian diets and related fractures. A prospective study of fracture risk in the United Kingdom found that fracture risk was higher in vegans as they had lower calcium intakes (<525 mg/day), but no difference was found between meat/fish eaters and lacto-ovo-vegetarians [90]. Vegan populations should obtain calcium from other sources—such as tofu, fortified soy products or fortified orange juice—but care is required to ensure an adequate intake. A careful selection of foods or the addition of fortified foods or supplements is needed to avoid a potential nutrient shortfall and to help ensure a healthy bone balance and reduce fracture risk in those individuals who adhere to vegetarian diets.

In the most recent meta-analysis published last year by Fabiani et al., the dietary pattern named as the “Meat/Western” pattern—characterized by a high consumption of red and processed meat, refined grains, and sweets—was positively associated with a lower BMD and a higher fracture risk. This dietary pattern was compared to “Healthy” patterns, characterized by a high consumption of fruit and vegetables, and the “Milk/Dairy” pattern, and these two patterns were associated with a decreased risk of low BMD. Moreover, the “Healthy” pattern had a significant preventive effect on fracture risk, whereas the “Meat/Western” pattern significantly increased the fracture risk [91]. A summary of larger studies relating bone status to each dietary pattern is found below in Table 2. 

Unfortunately, we have to point out that the majority of studies developed to elucidate the association between bone mineral status or fracture risk and dietary patterns had a cross-sectional design, and this fact may have influenced their results and further conclusions.

Interdependencies among nutrients could be a part of the explanation for the heterogeneity of the results from different studies and meta-analyses. These studies based on dietary patterns are very important for translating knowledge and facilitating recommendations for practice, and they should complement studies of single nutrients and particular foods in relation to bone health. Nutrients are consumed together in one meal rather than in isolation, and often, their effects will not be visible when the intakes of other nutrients are suboptimal.

## 4. Conclusions

In conclusion, the development of programs encouraging lifestyle changes (especially balanced nutrient intakes) is imperative for the reduction of osteoporosis risk, alongside regular physical activity.

The available evidence continues to raise questions that could be clarified with randomized controlled trials to establish the causal relationship between dairy consumption (as the major source of calcium, vitamin D and proteins) and the risk of fracture. Further research is needed. An adequate calcium intake seems to be recommendable for bone health, regardless of dietary preferences.

Overall, adherence to a healthy dietary pattern including fruit, vegetables, whole grains, poultry, fish, nuts and legumes, and low-fat dairy products and the avoidance of processed food products will be beneficial for bone health, decreasing the risks of osteoporosis and fractures.

## Figures and Tables

**Figure 1 nutrients-12-01986-f001:**
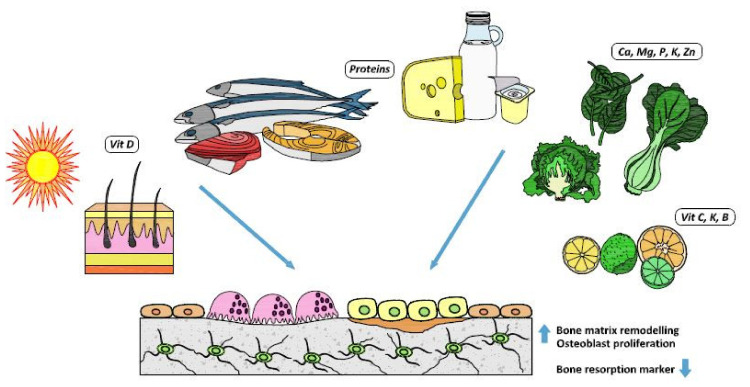
Nutrients and their bone effects.

**Table 1 nutrients-12-01986-t001:** Nutrient contents per 100 g of commonly used products.

Source	Calcium (mg)	Phosphorous (mg)	Potassium (mg)	Vitamin D (IU)	Protein (g)
**Whole fat milk**	119	93	151	*	3.3
**Skimmed milk**	122	101	156	*	3.4
**Swiss cheese**	791	567	77	20	26.9
**Cheddar cheese**	721	512	98	24	24.9
**Cream cheese**	98	106	138	*	5.9
**Yogurt low fat**	183	144	234	*	5.3
**Ice cream**	128	105	199	0	3.5
**Wild salmon**	9	0	360	600–1000	38
**Eggs**	56	197	138	20	12.5

Data adapted from [43]. * Not available, depending on the amount fortified in each country.

**Table 2 nutrients-12-01986-t002:** Summary of larger studies relating bone status to each dietary pattern.

Dietary Pattern	Name of the Study	No. Participants (Sex), Mean Age/Age Range	Effect on BMD	Effect on Fracture Risk
***Mediterranean Diet***	EPIC Study [61]	188,795 (139,981 women and 48,814 men) 48.6 years (±10.8)		7% decrease in hip fracture incidence
	CHANCES Project [62]	140,775 (116,176 women, 24,599 men) 60 years and older		4% decrease in hip fracture risk
	Meta-analysis, Malmir et al. [71]	358,746 13–80 years	Positive association with lumbar spine, femoral neck and total hip BMD	21% reduced risk of hip fracture
***Asian Diet***	Singapore Chinese Health Study [63]; vegetables–fruit–soy	63,257 (35,241 women and 27,913 men) 45–74 years		34% reduced risk of hip fracture
	Osteoporosis Korean Health and Nutrition Examination Survey 2008–10 [85]	3735 (postmenopausal Women) 64 ± 9 years	Dairy and fruit pattern decreased the risk of osteoporosis of the lumbar spine (53%); white rice, kimchi and seaweed dietary pattern was negatively associated with bone health	
***Western Diet***	Co-twin controlled study, United Kingdom [78]	4928 (postmenopausal Women) 56 ± 12 years	Inversely associated with BMD in femoral neck	
	Framingham Offspring Study, United States, [76]	2740 (1534 women and 1206 men) 29–86 years	Higher intakes of red meat and processed food were inversely associated with femoral neck BMD	
***Vegetarian Diet***	Bayesian meta-analysis [89]	2749 (1880 women and 869 men) 20–79 years	4% lower BMD (95% CI: 2%, 7%) at femoral neck and lumbar spine	
	EPIC Oxford study [90]	34,696 (26,749 women and 7947 men) 20–89 years		Fracture incidence rate ratios: 1.00 (0.89–1.13) for vegetarians and 1.30 (1.02–1.66) for vegans

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
