# Peer review of "Nutrients and Dietary Patterns Related to Osteoporosis"

_nutrients, 2020, doi:10.3390/nu12071986_

Round 1
Reviewer 1 Report
The review is interesting and cover most of the ralationships between nutritional elements, diet patterns and osteoporosis.
I would suggest to change the title, since diet "quality" has not been considered in the manuscrpit.
Moreover, I would suggest to split the section "Osteoporosisi and related nutrients" as well as "recebnt studies on reference dietary patterns" in several subsections, one for each individual nutrient as well as for each individual dietary pattern. This will improve readibility of the review.
Finally, I would suggest to merge the cells of dietary pattern in table 2 in single cells, one for each pattern
Author Response
Reviewer #1:
General comments
First, we want to thank you for your comments, which will undoubtedly contribute to improve the quality of our manuscript.
Specific comments
- The review is interesting and covers most of the relationships between nutritional elements, diet patterns and osteoporosis.
We appreciate your review of our article.
- 2. I would suggest to change the title, since diet "quality" has not been considered in the manuscript.
According to your suggestion we propose this new title “Nutrients and dietary patterns related to osteoporosis”
- Moreover, I would suggest to split the section "Osteoporosis and related nutrients" as well as "recent studies on reference dietary patterns" in several subsections, one for each individual nutrient as well as for each individual dietary pattern. This will improve readibility of the review.
As you recommend, we have made these changes to make the manuscript easier to read.
- Finally, I would suggest to merge the cells of dietary pattern in table 2 in single cells, one for each pattern
In the revised manuscript we have modified the table 2 following your recommendations.
Reviewer 2 Report
Please see my comments in the attached PDF file. This manuscript requires extensive English language editing.

Author Response
Reviewer #2:
General comments
First, we want to thank you for your comments, which will undoubtedly contribute to improve the quality of our manuscript. In response to your kind suggestions:
- This manuscript needs to be edited for serious English language problems.
According to your recommendation, the manuscript has been professionally edited for English language usage, grammar, spelling and punctuation to ensure clarity and readability and to conform to correct scientific English. The editor at Nutraceutical Translations has earned a PhD degree and has several years of writing, editing and publishing experience. The editor has undergone substantial editing training.
- Introduction. Line 39-40. “However, when dietary sources are not enough or not well tolerated, pharmacological calcium supplementation may be used”. Based on what evidence? What studies support supplementation of calcium? What studies do not support it?
This issue seems of special interest to us. According to Chiodini & Bolland (Eur J Endocrinol 2018), the dietary calcium intake may be adequate in most individuals, but, there is evidence supporting that in subjects with inadequate calcium and vitamin D intake, the supplementation strategies are useful for preventing osteoporosis-related fragility fractures. However, the different calcium intake among the different populations may be an important confounding factor in interpreting the results of the studies on the effect of calcium supplements on bone.
Overall, there are data suggesting that calcium supplements have positive effects on BMD, which is probably more important in subjects with adequate adherence to the supplements and with baseline lower dietary calcium intake (Kanis JA, et al, Osteoporosis International 2017). In addition, it should be taken into account that all available bone-active drugs are licensed in the context of an adequate calcium and vitamin D status, since the registration trials have been performed on patients always supplemented with calcium and vitamin D (Compston J, et al. Arch Osteoporosis, 2107). Nevertheless, some authors disagree with the usefulness of calcium and vitamin D supplements to prevent or treat osteoporosis due to their low efficacy and the possibility of adverse effects (Bolland MJ, Climacteric 2015).
Taking into account the previous considerations we have modified the paragraph in the introduction. “However, sometimes dietary sources are not enough or not well tolerated, and, in those situations, pharmacological calcium supplementation could be useful. This is the recommendation of most clinical practice guidelines. However, some authors are critical with this recommendation due to its low efficacy and the possibility of adverse effects.”
- Introduction. Line 39-40. “Which vegetables are highest in calcium? What about nuts and seeds?”
According to your suggestion we have modified the paragraph including the importance of nuts and seeds. Now we state that “The most important sources of calcium in the diet are dairy products (milk, yoghurt and cheese), fish (especially sardines with bones), pulses and a few vegetables and fruits (particularly nuts and seeds)”.
- Osteoporosis and related nutrients. Line 52. “antiostoporotic”
The typo has been corrected “anti-osteoporotic”
- Line 102. “bone markers of what”.
We have completed the sentence: “bone remodelling markers”.
- Line 340. “A systematic review about Meat Diet and Fish Diet”. “Do you mean dietary sources of meat and fish?”
Yes, we do. When we talked about Meat Diet we referred to diets with high consumption of meat or processed meat and when we said Fish Diet we referred to diets with high consumption of fish and seafood. We have modified this sentence in the text to be clearer as follows: “A systematic review about dietary patterns with a higher consumption of meat or processed meat (Meat Diet) or a higher consumption of fish and seafood (Fish Diet) suggested that these diets did not alter BMD or risk of fracture.” Thank you for pointing this out.
- Conclusions. Line 11-13. “Osteoporosis is a disabling disease that involves fragility fractures, high morbidity and mortality and high healthcare costs. Although drug therapy appears to be the first line option to reduce facture risk in the elderly is not always feasible and dietary modifications, specifically improving calcium, vitamin D and protein intakes, may be a more pragmatic option”. Reference? this information belongs in the introduction.
According to your suggestion we have included this paragraph in the introduction section and added the corresponding references:
“Osteoporosis is a disabling disease that involves fragility fractures, high morbidity and mortality and consistently higher healthcare costs”. “Although drug therapy appears to be the first line option to reduce facture risk in the elderly, it is not always feasible and dietary modifications, specifically improving calcium, vitamin D and protein intakes, may be a more pragmatic option”.
Round 2
Reviewer 2 Report
There is little improvement in English in this version. The abstract, in particular, is a nightmare of broken English. It is difficult to judge the quality of the article with such poor English, which makes the paper seem very amateurish. I strongly recommend the authors use the professional English editing service, specialist edit service.
Author Response
First, we want to thank you for your comments, which will undoubtedly contribute to improve the quality of our manuscript.
We agree with your comments regarding the English of the manuscript. For this reason, we have carried out an exhaustive review of English by 2 experts. We attach the editing certificate issued by the MDPI language editing service.
We hope that the improvements in English have made the manuscript more understandable and easier to read.
On the other hand, according to your suggestion, we have also rewritten the abstract to make it more comprehensible.
